# Nodding Syndrome: Clinical Characteristics, Risks Factors, Access to Treatment, and Perceptions in the Greater Mundri Area, South Sudan

**DOI:** 10.3390/pathogens12020190

**Published:** 2023-01-27

**Authors:** Gasim O. E. Abd-Elfarag, Jake D. Mathewson, Lukudu Emmanuel, Arthur W. D. Edridge, Stella van Beers, Mohamed B. Sebit, Robert Colebunders, Michaël B. van Hensbroek, Ente J. J. Rood

**Affiliations:** 1Amsterdam University Medical Centers, 1105 AZ Amsterdam, The Netherlands; 2Kit-Royal Tropical Institute, Epidemiology, Center for Applied Spatial Epidemiology (CASE), 1092 AD Amsterdam, The Netherlands; 3Access for Humanity (AFH), Monitoring and Evaluation Unit, Juba 81114, South Sudan; 4Department of Psychiatry, College of Medicine, University of Juba, P.O. Box 82, Juba 81114, South Sudan; 5Global Health Institute, University of Antwerp, Kinsbergen Centrum, Doornstraat 331, 2610 Antwerp, Belgium

**Keywords:** nodding syndrome, characteristics, risk factors, perceptions, South Sudan

## Abstract

We conducted a house-to-house survey in the Mundri, Western Equatoria state of South Sudan to investigate the clinical characteristics, risk factors, access to treatment and perceptions about nodding syndrome (NS). In total, 224 NS cases with median age of seizure onset of 10 years were identified. Head nodding only was reported in 50 (22.3%) cases, and head nodding plus other types of seizures in 174 (77.7%) cases. Wasting, stunted growth, delayed sexual development and speech and behavioral abnormalities were observed in 17 (23.6%), 16 (22.2%), 9 (17.3%), 14 (19.4%) and 4 (5.6%) cases, respectively. The consumption of rat meat, but not other bushmeat was associated with an increased risk of NS (OR 9.31, 95% CI 1.27–406.51). Children with NS were more likely to have taken ivermectin in the last 5 years (OR 2.40, 95% CI 1.33–4.43). NS cases were less likely to share a bedroom with other children (OR 0.06, 95% CI 0.02–0.16) or adults (OR 0.27, 95% CI 0.13–0.56). In conclusion, rat meat consumption is an unlikely risk factor for NS, and ivermectin intake was more common among NS cases than controls. Importantly, we documented that children with NS are stigmatized because of the misconception that NS is transmitted through direct contact.

## 1. Introduction

Nodding syndrome (NS) is an overwhelming neurodegenerative disease of unknown etiology. It has affected children and young adults from poor communities in multiple sub-Saharan Africa countries, including Tanzania, South Sudan, Uganda, the Democratic Republic of Congo (DRC), Cameroon, and the Central African Republic [1]. It often presents with distinct clinical features, including head nodding and other seizures that are often associated with debilitating complications, including impaired cognitive and physical development and delayed sexual maturity [2,3]. Previous attempts to identify a potential cause have focused on infections, nutritional deficiencies, toxins, autoimmune disease, hormonal and metabolic derangements, and genetic factors, but all were inconclusive [1]. Control of *Onchocerca volvulus* and its vector, the blackfly, by community-directed treatment with ivermectin (CDTI) and larviciding of rivers was proposed to prevent new cases of NS [4,5]. However, certain communities in the affected areas believe that NS is transmitted from person to person and therefore can be prevented by avoiding contact between cases and other family members or peers through eating and sleeping separately [1]. To further unravel the many unknowns around NS, we conducted a community-based study, investigating the clinical characteristics, potential NS risk factors, access to treatment and perceptions about NS transmission. Previously, we investigated epidemiological risks for NS at the household level, looking into socioeconomic status; source of water for drinking, cooking, and bathing; nutritional status; internal displacement; river distance; and CDTI. Ownership of poultry, behaviors around the rivers; and living near the rivers increased the risk of having NS in a household [6]. Here, we investigate the clinical characteristics of nodding syndrome (NS), individual-level risk factors, access to treatment, and perceptions about NS.

## 2. Materials and Methods

### 2.1. Study Setting 

This study was conducted in 13 selected bomas in the Greater Mundri area, Western Equatoria state of South Sudan (Figure 1). The Greater Mundri consists of 2 counties (Mundri West and Mundri East), which are sub-divided into 4 and 5 payams, respectively. Each payam is further sub-divided into an average of 4 bomas, consisting of individual villages. The population of the Greater Mundri was estimated in 2008 at 82,293 and was projected at 142,633 in 2020 [7,8]. Main livelihood activity includes subsistence farming, with fishing and animal husbandry as additional livelihood activities. Water is mostly accessed from boreholes, streams, and rivers. 

### 2.2. Study Design and Procedures 

From February 2018 to November 2019, we conducted an epidemiological study to determine the prevalence, distribution, and risk factors of NS; a clinical case–control study to determine the etiology and epidemiological risk factors of NS; and a follow-up study to investigate the disease progression and long-term outcome. Here we present data on the characteristics of NS, selected epidemiological risk factors, access to treatment, and perceptions about NS transmission. 

Epidemiological data were collected during a large house-to-house survey. Prior to initiating the household survey, community leaders in the area were informed about the aims and procedures of the study and involved in the planning and preparation of the study implementation in their communities. After obtaining informed consent, the main caretaker in a household was interviewed and all 1–18-year-old children in the household were screened for NS using head nodding as major inclusion criterion. If NS cases were identified in a household, questions on living conditions were asked. Every fifth household with an NS case and a control household without an NS case were asked additional questions regarding their individual exposures and behaviors. Household interviews were performed using electronic questionnaires preloaded on a personal digital assistant. The questionnaires were translated into the local language and translated back into English to ensure consistency. The study was conducted by a team of researchers consisting of a medical doctor, a clinical officer, nurses, field workers, and a field coordinator who had been trained prior to the survey. Screening of cases was conducted by the nurses while a clinical officer and/or a medical doctor carried out a detailed medical history and performed a clinical examination for confirmation. All the research team members were from the Greater Mundri area and were fluent in the local languages, including the local Arabic dialect. 

### 2.3. Data Quality 

After completion of household interviews, the field coordinator conducted data quality checks, including verifying personnel identifier code correctness, signing informed consent forms for household interviews and enrollment as cases/controls, and checking the data for completeness and accuracy and correcting inconsistencies. Once the data quality checks were completed, the data were extracted and uploaded directly from the Open Data Kit to the KoBoToolbox online database server for later analysis. Data were extracted from the online database and cleaned and de-duplicated before subsequent analysis. 

### 2.4. Statistical Analysis

Data were processed and ultimately analyzed using STATA version 15.1 (Stata Corp., College Station, TX, USA). Logistic regression with a logit link function was used to determine which covariate factors affect the probability of individuals being affected by NS, as opposed to the probability of not being affected (e.g., control). Individual data was matched to household-level data using unique household identifiers, which allowed for more extensive exploration of associations between cases and environmental factors recorded at household level. 

### 2.5. Ethical Consideration 

Ethical approval for the study protocol was attained from the ethics committee of the Ministry of Health of the Republic of South Sudan (approval date: 09-12-2016) and University of Antwerp, Belgium (reg.nr: B300201526244). Signed or thumb-print informed consent and assent was obtained from study participants or their parents or guardians. All study participants’ personal information was treated with strict confidentiality and coded for anonymity. 

## 3. Results

### 3.1. Clinical Characteristics of Nodding Syndrome

Overall, 22,411 persons were surveyed, of whom 607 (2.7%) were probable NS cases. Fifty (22.3%) presented with head nodding only and 174 (77.68%) with head nodding plus other types of seizures. The median age of onset of the nodding seizures was 10 years; 177 (94%) developed their first seizure between the ages of 3 and 18 years, and 10 (6%) between the ages of 1 and 2 years. In the majority (104/63.8%) of the NS cases, the onset of the seizures was in the 5–14-year-old age group (Figure 2). 

The most frequent type of seizure, in addition to head nodding, was generalized convulsions, reported in 160 (71.4%) of the NS cases (Table 1). In 116 (59.8%) and 108 (55.7%) cases, head nodding and other types of seizures were triggered by cold temperatures and the sight of food, respectively. Wasting, stunted growth, and delayed sexual development were observed in 17 (23.6%), 16 (22.2%), and 9 (17.3%), respectively. Speech and behavioral abnormalities were observed in 14 (19.4%) and 4 (5.6%), respectively. 

### 3.2. Risk Factors for Nodding Syndrome, Case–Control Study 

The results of the multivariate analysis for the effects of age, gender, consumption of bushmeat, and the use of ivermectin on the risk of NS are presented in Table 2.

Two hundred twenty-four cases and 238 controls were included for the case–control study. The prevalence of NS was higher in males than females (OR 1.28, 95% CI 0.97–1.69) but was not significantly associated with NS.

The consumption of rat meat increased the risk of NS (OR 9.31, 95% CI 1.27–406.51). However, consumption of other bushmeats including monkey, baboon, elephant, antelope, buffalo, squirrel, and bats was not significantly associated with NS. 

Prior use of ivermectin during the last 5 years increased the risk of NS (OR 2.40, 95% CI 1.33–4.43), yet the number of ivermectin doses (1 or 2) taken was not significantly different. 

### 3.3. Access to Treatment and Perceptions 

NS cases were reported to have had prior exposure to contacts with NS cases mostly at home (60%), at school (33%), and in the village (24%). However, NS cases were less likely to share a bedroom with other children (OR 0.06, 95% CI 0.02–0.16) or adults (OR 0.27, 95% CI 0.13–0.56) currently or in the past. 

Seventy-four percent of the NS cases received treatment for their condition, of which, 67% were through a doctor’s prescription, 33% were through self-medication, and 12% received traditional remedies. 

## 4. Discussion

The clinical characteristics of children with NS in the Mundri area were similar to NS cases reported from other areas. Firstly, their median age of seizure onset was 10 years, which is in line with previous studies [2,9,10]. Also, most recent NS cases were in the 5–14 years age group [11,12,13]. However, NS onset can occur in children younger than 5 years of age, as well as in individuals above 15 years of age, or in adults [3,14]. Secondly, in addition to head nodding seizures, generalized convulsions were the most frequent type of seizure, occurring in over 70% of cases, and the seizures were triggered by cold temperatures or the sight of food in over 50% of cases [2,12,14,15]. 

Concerning potential risk factors for NS, the consumption of rat meat was associated with increased risk of NS. However, the consumption of rats was only reported in 12.4% of the cases. Moreover, previous studies from South Sudan and northern Uganda did not find an association between NS and the consumption of other bush animals, including rodent brain, guinea fowl brain, and baboon meat and brain [16,17,18]. Of the known rodent diseases that can potentially transmit to humans, such as hantavirus pulmonary syndrome, hemorrhagic fevers, leptospirosis, lymphatic choriomeningitis, plague, rat-bite fever, salmonellosis, and other parasitic diseases, none is known to cause a progressive neurodegenerative disease such as NS [19,20]. In addition, rats are consumed as bushmeat not only by communities in the Greater Mundri area of Western Equatoria State but also by other communities in the Central and Eastern Equatoria states of South Sudan as one of their highly valued traditional foods, and unlike the Mundri area, NS does not exist in these communities. Therefore, the consumption of rat meat is unlikely to be the triggering factor of NS but may be confounded by poverty. 

A significantly higher proportion of NS cases were given ivermectin during CDTI as compared to the non-NS cases. Also, in a previous case–control study conducted in the same study area, 61.5% of NS cases received ivermectin at least once compared to 36.8% of the controls (OR 2.79, CI 0.64–11.75) [18]. A possible explanation for this may be that ivermectin is believed by the affected communities to improve the NS outcome. Also, a recent study in onchocerciasis-endemic areas in sub-Saharan Africa suggested that ivermectin intake reduced seizure frequency in persons with epilepsy who are infected with *O. volvulus* [21]. Additionally, children with NS, because of their disabilities, may be more likely to be at home during ivermectin distribution compared to healthy controls. Moreover, a study in Cameroon also suggested that annual CDTI is able to reduce the incidence of epilepsy in an onchocerciasis-endemic area [22]. Therefore, in South Sudan, there is a need for a longitudinal study with community-based surveillance for new NS cases and an improved CDTI program over a long period of time in order to confirm or refute such findings. 

Finally, we found that indeed there was prior exposure of the NS cases to contacts with NS, mostly at home but also at school and in the village. This, in addition to the fact that in the Mundri area NS cases are clustered in some families, has triggered beliefs among the community that NS is transmitted from person to person through the saliva, which further resulted in families isolating their NS cases from the healthy individuals by eating and sleeping separately. Similar beliefs and the behavior of isolating NS cases from other family members and peers as a preventive measure were also reported by studies from Uganda [23,24]. Furthermore, we also confirmed for the first time that NS cases are less likely to share a bedroom with other non-NS children or adults. These stigmatizing behaviors have significantly negatively impacted the NS-affected children, including increasing the likelihood of dropping out of school because of the fear of NS transmission from one child to another on school premises [23,25]. 

Our study had several limitations. Firstly, there was recall bias related to the medical history of the disease onset. Secondly, cases were identified by a clinical officer, nurses, and a medical doctor but not confirmed by a neurologist. Finally, our case–control study did not assess the role of *O. volvulus* infection as a risk factor of NS. However, several previous studies have reported a significantly strong association between NS and onchocerciasis [1]. The possible link of NS with the river has been confirmed by our most recent study in the Mundri area [6]. Also, a study in Maridi, in the Western Equatoria region of South Sudan, has reported that this link is with the blackfly breeding site at the Maridi dam [26]. Moreover, biological tests, including those for *O. volvulus*, were performed, but the results of these tests will be presented in an etiological case–control study. 

## 5. Conclusions

In conclusion, our study confirmed the previous clinical characteristics of NS and revealed that the majority of NS cases present with head nodding plus other types of seizures. Although we found an association of NS with the consumption of rat meat, given the epidemiology of NS, this is unlikely to play a role as a causal risk factor. More children with NS were reported to have had previous intake of ivermectin, but this most likely was the consequence of NS. Importantly, we documented that children with NS are stigmatized because of the misconception that NS is transmissible through direct contact. 

## Figures and Tables

**Figure 1 pathogens-12-00190-f001:**
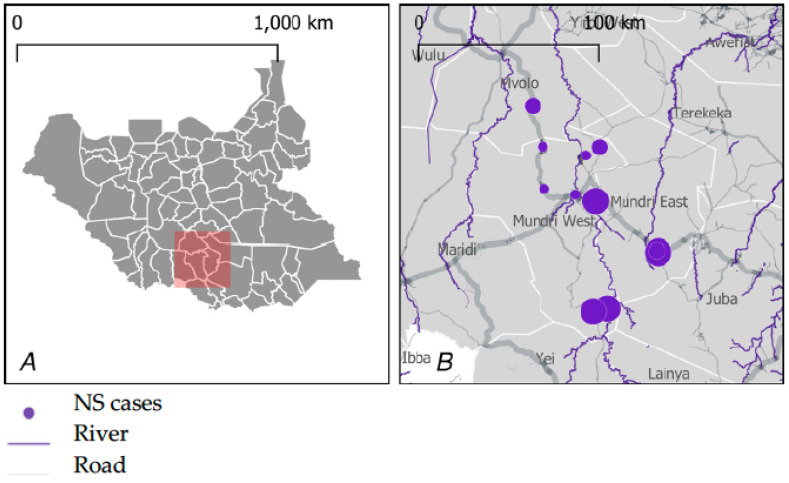
Study site general location (**A**) and distribution of NS cases in the study area (**B**). Figure adopted from Abd-Elfarag GOE et al. PLoS Negl Trop Dis 16(7): e0010630. https://doi.org/10.1371/journal.pntd.0010630 (accessed on 24 August 2022).

**Figure 2 pathogens-12-00190-f002:**
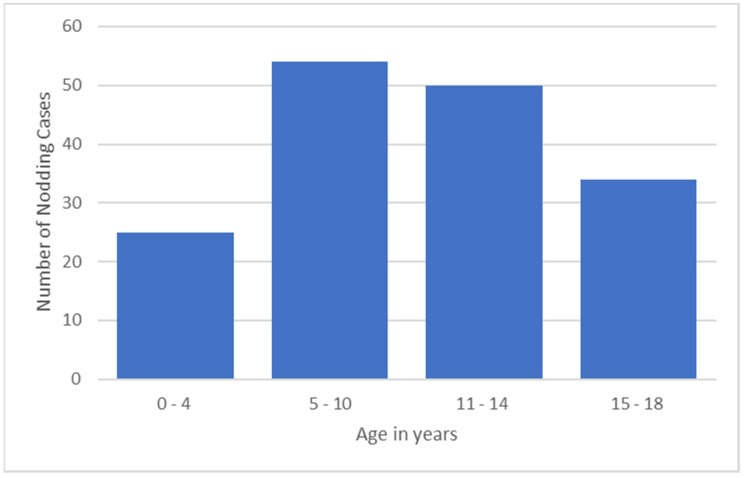
NS onset by age group.

**Table 1 pathogens-12-00190-t001:** Frequency of nodding syndrome morbidities in the Greater Mundri area, South Sudan.

Features	Number (%)
Head nodding	224 (100)
Generalized convulsions	160 (71.4)
Triggered by cold temperature	116 (59.8)
Triggered by the sight of food	108 (55.7)
Wasting	17 (23.6)
Stunted growth	16 (22.2)
Speech disorder	14 (19.4)
Absences	10 (13.9)
Delayed sexual development	9 (17.3)
Behavioral abnormalities	4 (5.6)

**Table 2 pathogens-12-00190-t002:** Risk of nodding syndrome between cases and controls by gender, consumption of bushmeat and intake of ivermectin in the Greater Mundri Area, South Sudan.

	Cases (n = 224)		Controls (n = 238)		OR (95% CI)	*p* Value
	#	%	#	%		
Male gender						
	110	49.6	129	54.7	1.28 (0.97–1.69)	0.07
Consumption of bushmeat						
Monkey	41	46.1	29	43.3	1.12 (0.56–2.23)	0.73
Baboon	34	38.2	22	32.8	1.26 (0.62–2.61)	0.49
Elephant	44	49.4	32	47.8	1.07 (0.54–2.12)	0.84
Antilope	65	73.0	53	79.1	0.72 (0.31–1.61)	0.38
Buffalo	53	59.5	44	65.7	0.77 (0.38–1.56)	0.44
Squirrels	48	53.9	39	58.2	0.84 ( 0.42–1.67)	0.59
Bats	5	5.6	1	1.5	3.92 (0.42–188.55)	0.18
Rat meat	11	12.4	1	1.5	9.31 (1.27–406.51)	0.01
Use of ivermectin						
Ivermectin intake during the last 5 years	36	64.3	514	42.8	2.40 (1.33–4.43)	0.002
Ivermectin doses received: 1	47	83.9	1,018	84.8	0.93 (0.44–2.21)	0.85
Ivermectin doses received: 2	9	16.1	182	15.2	1.07 (0.45–2.26)	0.85
Exposure and behavior						
Share bedroom with other children currently or in the pass	10	11.2	46	65.7	0.06 (0.02–0.16)	<0.001
Share a bedroom with adults currently or in the past	27	30.3	43	61.4	0.27 (0.13–0.56)	<0.001

## Data Availability

The datasets generated during the current study are available from the corresponding author on reasonable request.

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
