# Peer review of "Nodding Syndrome: Clinical Characteristics, Risks Factors, Access to Treatment, and Perceptions in the Greater Mundri Area, South Sudan"

_pathogens, 2023, doi:10.3390/pathogens12020190_

Round 1
Reviewer 1 Report
The manuscript titled "Nodding Syndrome clinical charecteristics...."submitted to Pathogen journal is good study and has good analysis. The manuscript describes about food behavior and associated neurological problem like Nod Syndrome and epilepsy/seizure. It is worth to consider for publication. But I would have some questions and if those answers included in manuscript , then it will be much informative about Nod Syndrome.
Is there any trasmission risk of O.VULVULUS THAT CAUSES SEIZURE OR EPILEPSY IN RAT MEAT PATIENTS?
WHAT IS THE ROLE OF IMINECTIN IN RAT MEAT PATIENTS?
IS THERE ANY INCIDENCE OF NS IF RAT MEAT IS ASSOCIATED WITH BUSH MEAT?
IS THERE ANY SPECIFIC CAUSE RAT MEST LEAD TO Nod Syndrome(NS).
Reviewer 2 Report
A well written, easy-to-ready article.
The article focuses on nodding syndrome (NS), defined as an involuntarily nodding of the head. The article examines the correlation between NS and the presence of other clinical signs and behaviors. The article finds an association with generalized convulsions (71% of cases) and demonstrates with statistical confidence that healthy individuals avoid sleeping in the same room as those afflicted with NS (OR < 0.1) presumably because of the belief that NS is transmitted by contact. The article also finds a positive association between NS and eating rat-meat (OR = 9.3). While rat-meat consumption increases the odds of having NS, it is unlikely to be the primary cause for NS as it is found in only 12% of the diseased individuals.
The main “deficiency” of the article is a failure to closely examine O. volvulus infection as a risk factor. This is surprising because one of the authors has been a strong proponent that NS is linked to onchocerciasis. In fact, the team deliberately selected an onchocerciasis-endemic area to enroll participants. It would therefore have been interesting to look for associations with clinical signs of onchocerciasis (or serology). Perhaps the authors are planning on doing so in a separate study. Regardless, it would help if the authors were to explain why onchocerciasis was not investigated more in depth (beyond ivermectin intake in MDA).
The main suggestion for improvement is to examine the correlation between prevalence of NS and distance from a fast-flowing river, as a proxy for risk of onchocerciasis. Figure 1 B suggests that the prevalence is higher close to rivers, and this might be quantified, and turned into an odd ratio.
Finally, there is a copy/paste error: Lines 71-75 appear to belong to a template and should be removed.
